

# The effect of diluted 1% baby shampoo on biofilm reduction in chronic rhinosinusitis with nasal polyposis

Farah Dayana Zahedi[1], Mun Yee Soo[2], Muttaqillah Najihan Abdul Samat[3], Suria Hayati Md Pauzi[4], Salina Husain[1], Aneeza Khairiyah Wan Hamizan[1], Baharudin Abdullah[5], Balwant Singh Gendeh[6], Abdul Samat Ismail[7] and Rocky Vester Richmond[7]

[1] Department of Otorhinolaryngology-Head and Neck Surgery, Faculty of Medicine, Universiti Kebangsaan Malaysia, Cheras, Kuala Lumpur, Malaysia

[2] Department of Otorhinolaryngology-Head and Neck Surgery, Hospital Tuanku Azizah, Kuala Lumpur, Malaysia

[3] Department of Microbiology, Faculty of Medicine, Universiti Kebangsaan Malaysia, Cheras, Kuala Lumpur, Malaysia

[4] Department of Pathology, Faculty of Medicine, Universiti Kebangsaan Malaysia, Cheras, Kuala Lumpur, Malaysia

[5] Department of Otorhinolaryngology-Head and Neck Surgery, School of Medical Sciences, Universiti Sains Malaysia, Kota Bharu, Kelantan, Malaysia

[6] Department of Otorhinolaryngology-Head and Neck Surgery, Pantai Hospital Kuala Lumpur, Kuala Lumpur, Malaysia

[7] Department of Paediatrics, Faculty of Medicine, Universiti Kebangsaan Malaysia, Cheras, Kuala Lumpur, Malaysia

Corresponding author
Farah Dayana Zahedi,
farahdayana@ukm.edu.my

## ABSTRACT

**Background.** Biofilm has been identified as the contributing factor for refractory chronic rhinosinusitis with nasal polyposis (CRSwNP). Nasal douching using baby shampoo was thought to be effective in patients with CRSwNP. We aimed to study the *in-vitro* reduction of biofilm using diluted 1% baby shampoo.

**Methods.** Sixty nasal polyps taken from patients who met the inclusion and exclusion criteria were sent for histopathological examination using haematoxylin and eosin staining. Another portion of the same samples was sent for tissue culture and tissue culture plate assay to identify *S. aureus* and *P. aeruginosa* and determine their biofilm forming capacity. The efficacy of diluted 1% baby shampoo versus normal saline was tested on the biofilm *in vitro* where the optical density readings were compared pre- and post-treatment.

**Results.** The prevalence of biofilm in patients with CRSwNP is 21.7%. Thirteen samples were positive for biofilm; *P. aeruginosa* 23% ($n = 3$), *S. aureus* 15% ($n = 2$), no bacterial growth 54% ($n = 7$) and others 8% ($n = 1$). Biofilm formation was significant in both *S. aureus* and *P. aeruginosa* ($p < 0.001$) whilst a significant reduction of biofilm was seen in diluted 1% baby shampoo ($p = 0.043$).

**Conclusion.** In conclusion, diluted 1% baby shampoo is an effective treatment in the reduction of biofilm for CRSwNP.

## INTRODUCTION

Chronic rhinosinusitis (CRS) is characterized by mucosal inflammation of the nose and paranasal sinuses with sinonasal symptoms which persist for more than 12 weeks (*Cain & Lal, 2013*). CRS is a polymicrobial disease with a wide range of pathogens involved. It was difficult to determine which pathogens were predominant due to various sampling techniques (*e.g.*, swab, biopsy, irrigation, and aspiration), the inability to maintain sterility as the nasoendoscope was passed through, and different methods of culture (*Meltzer et al., 2004*). Pathogens which are commonly found in patients with chronic rhinosinusitis with nasal polyposis (CRSwNP) were *S. aureus, H. influenzae, P. aeruginosa* and fungus (*Smith, Buchinsky & Post, 2011*). According to a study done by *Zahedi et al. (2019)*, commonest pathogens cultured from swabs taken from middle meatus were *Pseudomonas* sp., *Enterobacter* sp. and followed by coagulase-negative staphylococcus (CONS).

Biofilm is an organized, heterogeneous bacterial community embedded in extracellular polymeric substances (EPS), which are predominantly water and rich in polysaccharides, nucleic acid and proteins (*Zhao, Sun & Liu, 2023*). It is a complex three-dimensional structure which contains either single species or different species of microbial, organized in patches of separate colonies and each sub-specialized into different functions (*Costerton et al., 1995*). The free-floating planktonic bacteria adhered to a biological surface to form microcolonies which eventually progressed to form a biofilm. Over the years, studies have revealed the possibility of bacterial biofilm formation, which was postulated to be the driving cause of the disease being refractory to treatment (*Tomooka, Murphy & Davidson, 2000*). Biofilms were more commonly found in patients with chronic rhinosinusitis with nasal polyposis. Studies have shown that CRSwNP patients who failed maximum medical therapy and surgery, had a biofilm positive rate of 20–100% (*Li et al., 2012*).

Nasal irrigation is one of a few treatment options used for CRSwNP. Various studies have been carried out to compare different substances used for nasal irrigation such as normal saline, antibiotic, steroid, and surfactants (*Adappa, Wei & Palmer, 2012*; *Turner et al., 2017*). Normal saline nasal irrigation has been popularized over the years and achieve worldwide acceptance as an adjunctive therapy (*Macassey & Dawes, 2008*). A promising result from a study reported that diluted 1% baby shampoo improved nasal symptoms and scope findings in a group of patients following surgery, generating further interest of diluted 1% baby shampoo nasal irrigation as a treatment option (*Chiu et al., 2008*). Diluted 1% baby shampoo serves as a biological and chemical surfactant with antimicrobial potential. It disrupts cell membranes, increases membrane permeability leading to potential metabolite leakage, and interferes with membrane functions (*Van Hamme, Singh & Ward, 2006*).

Various imaging modalities have been used in different studies to detect the presence of biofilms. Imaging modalities include scanning and transmission electron microscopy, fluorescent *in situ* hybridization (FISH) and confocal scanning laser microscopy (CSLM) (*Vuotto & Donelli, 2014*; *Nistico et al., 2009*; *Schlafer & Meyer, 2017*). Detection of biofilms using hematoxylin and eosin (H&E) staining was first reported in 2010 by *Hochstim et al. (2010)*; *Tóth et al. (2011)* also reported that gram stain had a strong correlation with H&E

staining and was a reliable predictor of the presence or absence of biofilm. There are a few microbiological methods used for biofilm detections *in vitro*. These include Tissue Culture Plate (TCP), Tube Method (TM) and Congo Red Agar (CRA). Among the three methods, Tissue Culture Plate has been demonstrated as the most sensitive and specific in detecting biofilms (*Deka, 2014*).

Many previous *in-vitro* studies explored the efficacy of surfactants in inhibiting the biofilm formation, such as using citric acid/zwitterionic surfactant (CAZS), SinuSurf, and baby shampoo. But from all the studies, none was done on nasal polyps' specimens taken from CRSwNP patients (*Desrosiers, Myntti & James, 2007*; *Chiu et al., 2008*; *Kofonow & Adappa, 2012*). A study done by *Chiu et al. (2008)*, reported that Johnson & Johnson baby shampoo of 1% concentration was effective in inhibiting the formation of biofilm with favourable clinical improvements as compared to other concentration, thus making it as a determined concentration for clinical study. Although the study reported the efficacy of the 1% concentration baby shampoo in inhibiting the biofilm formation in mostly CRS, there was lack of data exploring this effect specifically on patient with CRSwNP.

Hence, the aims of this study were to determine the prevalence of biofilm, investigate the biofilm forming capacity of *Staphylococcus aureus* and *Pseudomonas aeruginosa* and define the effect of diluted 1% baby shampoo on biofilm reduction using nasal polyp specimens of patients with CRSwNP. To the best of our knowledge, this is the first study conducted to compare the effects of diluted 1% baby shampoo in reducing biofilms *in vitro* on nasal polyp specimens of patients who underwent endoscopic sinus surgery (ESS) for CRSwNP.

## MATERIALS & METHODS

### Study population

This study received the Research Ethical Committee, Universiti Kebangsaan Malaysia (REC UKM) approval with the approval number of JEP-2016-510. All patients underwent ESS for CRSwNP within the study period of 2 years who fulfilled both the inclusion and exclusion criteria were recruited into the study. The inclusion criteria were age 18 years old and above, who underwent ESS for CRSwNP. Patients who have sinonasal tumour and/or diagnosed with granulomatous nasal diseases were excluded from the study. Written informed consent in accordance to institutional guidelines was taken from patients who agreed to be involved in this study.

### Methodology

Two sections of the samples of nasal polyp were obtained from each patient and samples were sent to the histopathology (for one section) and microbiology lab (for another section) as per laboratory protocols on the same day (Fig. 1).

### Histopathology lab methology

*H&E on Biofilm.* One section was stained with H&E as protocol modified from *Hochstim et al. (2010)*. The section was deparaffinized in xylene twice for 5 min (2 × 5 min) and rehydrated with successive 1-minute washes in 100%, 96%, 80%, and 70% ethanol. Subsequently, the specimen was stained with haematoxylin for two minutes, rinsed with

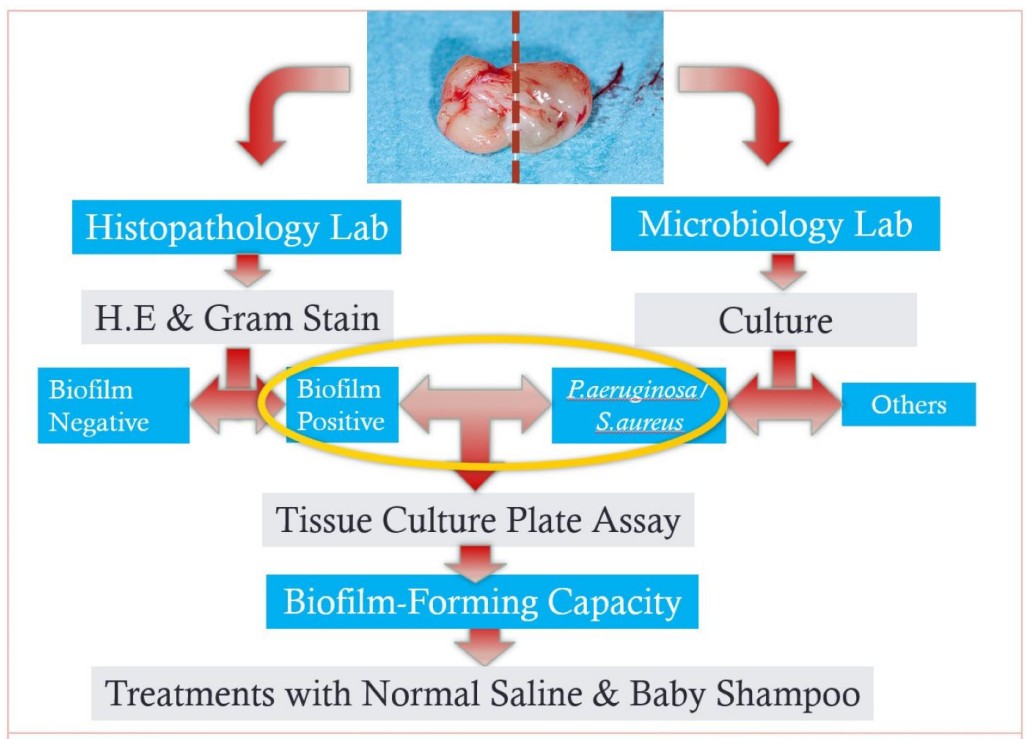

**Figure 1** **A flowchart of methodology from sampling to *in-vitro* tests.**

distilled water, rinsed with 0.1% hydrochloric acid in 50% ethanol, rinsed with tap water for 15 min, stained with eosin for one minute, and rinsed again with distilled water. The slide was dehydrated with 95% and 100% ethanol, successively followed by xylene twice for five minutes ($2 \times 5$ min) and mounted with coverslips.

*Biofilm identification for H&E.* Slides were examined and evaluated by two researchers. Presence of irregularly shaped groupings of small basophilic bacterial clusters, one third of the size of the surrounding epithelial or inflammatory cells, a biofilm seen over the epithelial lining, not in or under the epithelial lining, biofilm seen tightly adherent to the surface epithelium or pulled away slightly; or a dense extracellular polysaccharide substance (EPS) material with embedded basophilic bacteria, occasionally entrapped erythrocytes, leukocytes and partially sheared from epithelial surface substance coating the epithelial surface, indicating biofilm-positive (*Tóth et al., 2011*) (Fig. 2).

*Gram staining of the samples.* Gram staining of the samples following the protocols of Gram stain kit Bio-Optica 04-100802 was employed to complement the histopathological examination (HPE) findings of biofilm (*Mangels, Cox & Lindberg, 1984*, *Bortholomew, 1962*).

*Biofilm identification criteria.* Histological criteria to denote the presence of biofilms were the morphologic features of biofilms on H&E staining as mentioned above, Gram positive or

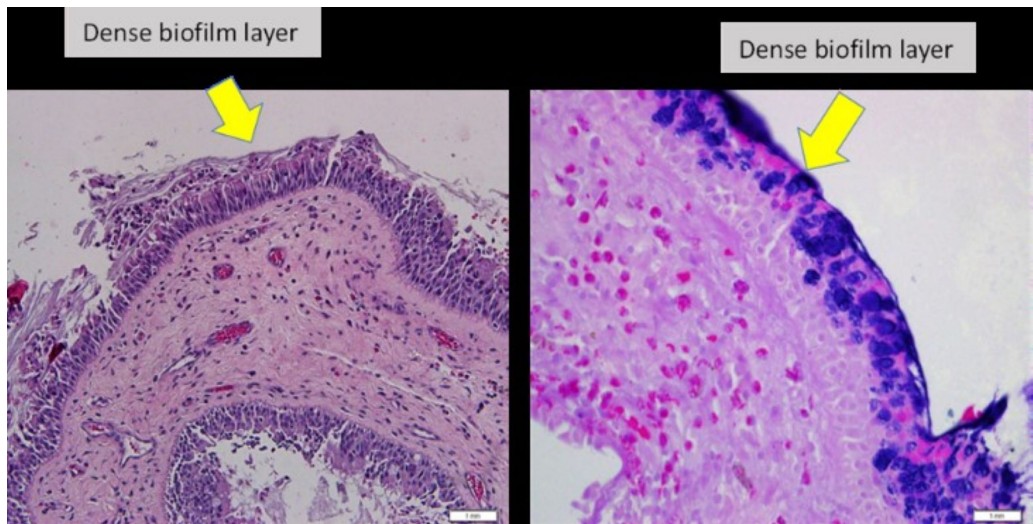

**Figure 2  Nasal polyp histopathology specimen.** (Left) Nasal polyp specimen with H&E staining, under light microscopy 20x magnification. (Right) Nasal polyp specimen with Gram staining, under light microscopy 40x magnification.

Gram negative and microcolonies seen in Gram staining, and the presence of surrounding polysaccharide layer.

## Microbiology lab methodology

*Tissue culture plate assay.* Another section of the tissue specimens obtained from the operating theatre were immediately transported to the microbiology lab. The tissue culture plate assay employed in this study was based on the method described previously by *Ha et al. (2008)*. The tissue specimens were homogenized. Specimens were inoculated on blood agar, MacConkey agar, mannitol salt agar and cetrimide selective agar, and incubated at 37 °C for 24 h. Pure colonies from the isolates above were then inoculated on to nutrient agars until further testing. Yellow colonies grew on mannitol salt agar were subjected to deoxyribonuclease (DNase) and coagulase test. DNase positive and coagulase positive colonies were considered as *S. aureus*. Green-pigmented colonies which grew on cetrimide agar were subjected to the API 20 NE test to confirm the identification of *P. aeruginosa*. Three to four colonies of bacterial isolates were suspended in Mueller-Hinton broth until the turbidity matches a standard of 0.5 McFarland and were incubated aerobically at 37 °C for 24 h. One mL of the bacterial suspension was diluted 1:100 with fresh Mueller-Hinton broth ($10^6$ colony forming unit (CFU) mL). One mL of this 1:100 bacterial suspension was added to 1ml of sterile Mueller-Hinton broth to achieve a final concentration of $5 \times 10^5$ CFU/mL. Reference strain of biofilm-forming *S. aureus* ATCC 25923 and *P. aeruginosa* ATCC 27853 obtained from the American Tissue Culture Collection (ATCC) were used as a positive control in this study. Control strains were cultured with the same method. Biofilm presence of the tissue specimen was confirmed using H&E and Gram staining. Only tissues with biofilm and culture positive for *S. aureus* or/and *P. aeruginosa* were included in the analysis. The organisms were tested for biofilm-forming properties.
Three 96-well tissue-culture treated microplates were filled with 200 µL of Mueller-Hinton broth and 10 µL of the final inoculums ($5 \times 10^5$ CFU/mL) and were incubated at 37 °C for 8 days without agitation for biofilm growth. 50 µL of media were removed from each well and replaced with 50–150 µL of fresh media on alternate days to achieve maximal biofilm growth. After incubation for 8 days, the wells were washed twice with distilled water to remove planktonic forms of bacteria and were left to dry. One of the 96-well tissue-culture treated microplates was filled with 250 µL of methanol and left for 15 min for fixation of adherent bacteria. Subsequently methanol was removed, and the microplate was air dried for another 15 min. Subsequently 200 µL of 0.1% crystal violet (CV) solution was applied to stain the adherent bacteria for 5 min. Then, the CV solution was decanted, and the wells were washed three times with distilled water. The stained wells were filled with 250 µL of 95% ethanol and were incubated for 1 h on a rocking platform at room temperature to solubilize the adherent material.

*Quantification of biofilm formation*. Quantification of biofilm growth was determined by reading the optical density (OD) of each well at 595 nm (OD595nm) using a microplate reader. Reference strains were used as control. Negative control wells were filled with only sterile broth. Test was carried out in a triplicate manner, and the mean OD was recorded. A cut-off optical density (ODc) needs to be established to note the presence or absence of biofilm. The cut-off optical density (ODc) was defined as 3 standard deviations above the mean OD of the negative control (culture medium): ODc=average OD of negative control+($3 \times$SD of negative control). The mean OD of the strains will be compared to the OD of the negative control. Analysis was done as described by *Stepanović et al. (2007)*, whereby strains were classified as follows: OD $\leq$ ODc no biofilm producer, ODc<OD $\leq$ $2 \times$ ODc weak biofilm producer, $2 \times$ ODc<OD $\leq$ $4 \times$ ODc moderate biofilm producer and $4 \times$ ODc<OD strong biofilm producer.

### Preparation of diluted 1% baby shampoo dilution

An amount of 0.1 ml of Johnson & Johnson baby shampoo was diluted in 10 mls of normal saline to achieve a 1% concentration which was used as nasal irrigation (*Joss et al., 2016*). This concentration was determined and adapted based on previous literature (*Chiu et al., 2008*). Solution was prepared 30 min prior to test.

### Biofilm reduction testing

Once the biofilm had been established and confirmed on the first microplate on day 8, wells on the second and third microplates containing the same inoculums were filled with 100 µL of 1% baby shampoo diluted in normal saline, and normal saline separately. The microplates were incubated for another 24 h at 37 °C. Positive control wells contained only the inoculums and the media without the test component, and negative control wells contained only test component and media, without the inoculums. The microplates were analysed using a crystal violet assay and optical density and measured with a microplate reader as described earlier. A flow chart of the methodology is shown in Fig. 1. All tests were carried out in a triplicate fashion. The relative inhibition of biofilm was calculated as

**Table 1  Bacteria isolates cultured from biofilm-positive samples.**

| Bacteria | Frequency (*n*) | Percentage (%) |
|---|---|---|
| No Growth | 7 | 54 |
| *S. aureus* | 2 | 15 |
| *P. aeruginosa* | 3 | 23 |
| Others | 1 | 8 |

described by *Ha et al. (2008)*, as follows:

**Percentage of biofilm inhibition** $= 100 - [OD_{595}$ of treated well$/OD_{595}$ of untreated well$]$ $\times 100$.

## Statistical analysis

Data was entered and analysed using SPSS 25.0. Wilcoxon signed-rank test was employed to analyse the reduction of biofilm using diluted 1% baby shampoo and normal saline only, and to compare biofilm reduction between normal saline and diluted 1% baby shampoo. Fisher's exact test was used to determine the statistical difference in biofilm formation and biofilm forming capacity of *S. aureus* and *P. aeruginosa*. $p < 0.05$ was considered as statistically significant.

## RESULTS

### Demographic data

A total of 60 samples were collected from 30 patients who underwent ESS for CRSwNP. The age ranged from 26 to 78 years, with a mean $\pm$ standard deviation (SD) of $58 \pm 15$ years. There were 21 male and nine female which comprised of 70% and 30% respectively, and 13 (43.3%) of them were Malays, 13 (43.3%) were Chinese and four (13.3%) were Indian. All patients were treated with intranasal corticosteroids and normal saline nasal douche. Half of the population underwent repeated ESS.

### Data on biofilms

Biofilms were detected in 13 out of 60 samples (21.7%) using H&E complimented with Gram stain method (Fig. 2). Out of these 13 samples, 23% ($n = 3$) comprised of *P. aeruginosa*, 15% ($n = 2$) of *S. aureus*, 54% ($n = 7$) had no bacterial growth and 8% ($n = 1$) belonged to other bacteria (Table 1). All samples which grew *S. aureus* ($n = 2$) or *P. aeruginosa* ($n = 3$) were bio-film-positive. There was no sample which grew *S. aureus* and *P. aeruginosa* simultaneously. There was statistical significance in biofilm formation of *S. aureus* or *P. aeruginosa* with the $p$-values of 0.0002 ($p < 0.05$) (Table 2).

### Biofilm forming capacity

All samples which grew *S. aureus* or *P. aeruginosa* with biofilm seen on H&E staining were subjected for tissue culture plate assay to determine the bacteria biofilm forming

**Table 2  Distribution of organisms forming biofilm.**

|  | Biofilm | No Biofilm |  |
| --- | --- | --- | --- |
| *S. aureus/P. aeruginosa* | 5 | 0 | 5 |
| Others | 8 | 47 | 55 |
|  |  | Total | 60 |
|  |  | *p*-value | 0.0002 |

**Table 3  Comparison of biofilm reduction between normal saline and diluted 1% baby shampoo.**

|  | Untreated (OD) | Treated with Normal Saline (OD) | Reduction with Normal Saline (OD) | %Reduction with Normal Saline |
| --- | --- | --- | --- | --- |
| N | 5 | 5 | 5 | 5 |
| Mean | 3.02 | 2.15 | 0.90 | 26.87 |
| Median | 3.46 | 2.11 | 1.37 | 30.35 |
| Std. Dev | 1.38 | 1.06 | 0.72 | 23.46 |

|  | Untreated (OD) | Treated with 1% Baby Shampoo (OD) | Reduction with 1% Baby shampoo (OD) | %Reduction with 1% Baby shampoo |
| --- | --- | --- | --- | --- |
| N | 5 | 5 | 5 | 5 |
| Mean | 3.02 | 1.15 | 1.87 | 57.31 |
| Median | 3.46 | 1.35 | 2.02 | 58.41 |
| Std. Dev | 1.38 | 0.51 | 1.07 | 18.96 |

capacity. They were all graded after a cut-off point of optical density of the negative control was determined. The biofilm forming capacity categorizes the isolates into either a non-, weak-, moderate-, or strong biofilm producer, based on the optical density reading as described in the methodology. All samples with *P. aeruginosa* (100%) isolates were strong biofilm forming capacity, whereas one sample with *S. aureus* had strong biofilm forming capacity and one weak biofilm forming capacity (50%). There was no statistical difference in biofilm forming strength between *S. aureus* and *P. aeruginosa* with the *p*-value of 0.4.

## Effects of diluted 1% baby shampoo on biofilm mass

The five samples with biofilms (three of *P. aeruginosa* and two of *S. aureus*) were then tested for biofilm reduction using normal saline alone, and with a diluted 1% baby shampoo. The mean reduction in biofilm was 26.87% with normal saline alone and 57.31% after treatment with diluted 1% baby shampoo (Table 3). There was significant reduction of biofilm with the treatment of diluted 1% baby shampoo with the *p*-value of 0.043, however there was no significant reduction of biofilm with normal saline with the *p*-value of 0.080 (Table 4). Overall analysis diluted 1% baby shampoo was significantly more effective in biofilm reduction than normal saline with the *p*-value of 0.043 ($p < 0.05$) (Table 4, Fig. 3).

**Table 4** Effectiveness of biofilm reduction between normal saline and diluted 1% baby shampoo.

|  |  | N | Mean rank | Sum of ranks | Z | P |
|---|---|---|---|---|---|---|
| Normal saline | Negative ranks | 4 | 3.50 | 14.00 | −1.753 | 0.080 |
|  | Positive ranks | 1 | 1.00 | 1.00 |  |  |
|  | Ties | 0 |  |  |  |  |
|  | Total | 5 |  |  |  |  |
| 1% Baby shampoo | Negative ranks | 5 | 3.00 | 15.00 | −2.023 | 0.043 |
|  | Positive ranks | 0 | 0.00 | 0.00 |  |  |
|  | Ties | 0 |  |  |  |  |

**Notes.**

$N$, number of observations/samples; $Z$, $Z$-value; $P$, $P$-value.

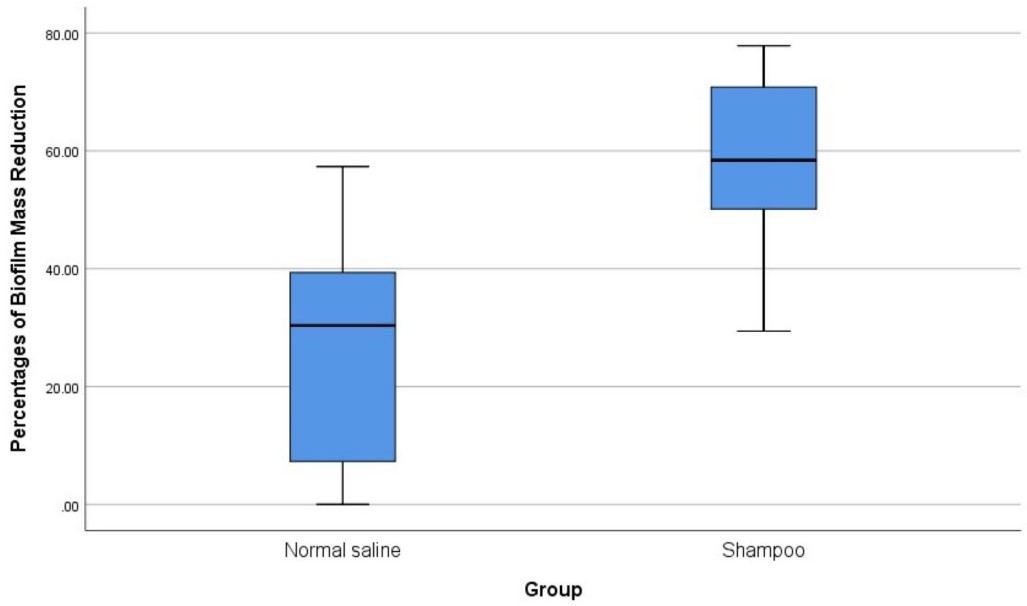

**Figure 3** Comparison of the effects of normal saline and diluted 1% baby shampoo treatment in biofilm mass reduction.

# DISCUSSION

Chronic rhinosinusitis has been reported to affect quality-of-life more than those suffering from chronic obstructive pulmonary disease, congestive heart failure, ischemic heart disease and back pain (*Metson & Gliklich, 2000*). It remains to be a challenging disease in terms of treatment whereby disease was inadequately controlled despite a combination of maximal medical therapy and ESS (*Hong et al., 2014*). One of many postulations for the chronicity of the disease is the involvement of bacterial biofilms, which made the planktonic bacteria resistant to the conventional treatment strategies (*Tomooka, Murphy & Davidson, 2000*).

The prevalence of biofilm in CRSwNP in our study is 22%, detected using a combination of H&E and Gram stain method. Morphological features observed in all biofilms positive

specimens in this study were consistent with the findings reported by *Hong et al. (2014)*, which are irregular groupings of small basophilic bacterial clusters, presence of a biofilm over the epithelial lining, biofilm tightly adherent on the surface epithelium and a dense extracellular polysaccharide substance with embedded basophilic bacteria. There is no standard methodology in biofilms detection ranging from light microscopy to crystal violet tissue culture plate assay. Hence the wide range of accuracy in biofilm-positive reports were observed in previous studies of patients with failure of maximal medical therapy and ESS (*Li et al., 2012*). Among the commonly used imaging modalities in biofilm studies are scanning electron microscopy (SEM) and transmission electron microscopy (TEM). The advantages over H&E method are that these methods able to show the structure, developmental stages and polymicrobial nature of biofilms, but the disadvantages are the difficulty in fixation process and in speciating individual bacterial species. Currently, confocal laser scanning microscopy (CLSM) is the most used technique in biofilm studies, as it provides a three-dimensional view of biofilm structures and also able to speciate the bacteria visualized using Bac light technique (*Suh, Cohen & Palmer, 2010*). However, CSLM is time-consuming, incurs much higher costs and not readily available.

There were 15% of *S. aureus* biofilm and 23% of *P. aeruginosa* biofilm detected in this study. It is also evidenced in this study that *S. aureus* and *P. aeruginosa* were significantly associated with biofilm formation. This is consistent with previous studies which demonstrated that *S. aureus* and *P. aeruginosa* were predominantly found in biofilms positive specimens of patients with CRSwNP. In a study of 157 patients, *S. aureus* and *P. aeruginosa* found in 71% of samples which showed biofilm growth (*Hong et al., 2014*). *Bendouah et al. (2006)*, reported 14 out of 19 (73%) patients had isolates of *S. aureus* or *P. aeruginosa* biofilm. Many times the individual or mixed species of bacteria or fungi which formed the biofilms were unable to be isolated or cultured by the conventional microbiological methods; hence the diagnosis is usually uncertain (*Tóth et al., 2011*). This is because the bacteria in the biofilms have significant low metabolic rate which renders the low culturability (*Suh, Cohen & Palmer, 2010*). Studies have reported there were significant associations of biofilms with nasal polyps. However, there was another group of authors reported no associations of biofilms and nasal polyps (*Karosi et al., 2013*; *Tóth et al., 2013*; *Sun et al., 2012*; *Bezerra et al., 2011*; *Mladina & Skitarelić, 2010*; *Mladina & Skitarelić, 2010*). There were studies which reported the heterogeneity of microbial community at different locations in the nasal cavity which demonstrated approximately 25% of microbiome variations within-patient in CRS (*Joss et al., 2016*; *Biswas et al., 2015*; *Yan et al., 2013*).

Tissue culture plate assay with crystal violet was used for *in vitro* study of biofilms on nasal mucosa specimens from patients with CRSwNP whereby OD of the wells which biofilms were read using TECAN Infinite F50 microplate reader at the wavelength of 590 nm. In our study, the cut-off OD for *S. aureus* and *P. aeruginosa* was 0.72 and 0.74, respectively. The OD of the reference strain of biofilm-forming *S. aureus* ATCC 23923 and *P. aeruginosa* ATCC 27852 which were used as positive controls were 3.97 and 1.69 respectively. Although we observed that all *P. aeruginosa* biofilm had strong biofilm forming capacity as compared to *S. aureus*, which demonstrated only one out of two has

strong biofilm forming capacity, we were unable to prove the statistical difference in the strength of biofilm forming capacity between these two organisms. This was most likely attributed to the low number of sample size which were positive for biofilm forming *S. aureus* and *P. aeruginosa*. A study done by *Prince et al. (2008)* reported that *P. aeruginosa* has higher propensity to biofilm formation as compared to the other bacteria found in the study such as *S. aureus*, CONS and H. influenza. It is also reported that *P. aeruginosa* is an organism with strong biofilm forming capacity which is consistent with our results (*Prince et al., 2008*). Conversely, *Foreman & Wormald (2010)*, reported that *S. aureus* has higher propensity for biofilm formation as compared to *P. aeruginosa*. A study done by *Bendouah et al. (2006)* reported that the presence of biofilm forming *S. aureus* or *P. aeruginosa* also predicts an unfavorable outcome of ESS.

Due to the recalcitrant nature of the disease, adjunctive therapy with topical application of various substances had been used and studied besides the mainstay treatment of intranasal and oral corticosteroid. The use of topical antibiotics may serve as an adjunctive treatment option for refractory CRS without nasal polyp, however, their use is not supported for CRSwNP (*Griffin et al., 2018*). Xylitol nasal irrigation has bactericidal properties which may inhibit biofilm formation (*Lin et al., 2017*). In our study, we compared the reduction of biofilm mass *in vitro* using diluted 1% baby shampoo and normal saline alone on nasal mucosa specimens. We observed that diluted 1% baby shampoo showed significant reduction of biofilm mass of nasal mucosa specimens as compared to normal saline. Baby shampoo serves as a surfactant with both biological and chemical effects on respiratory mucosa. Its mucolytic effect has been demonstrated to decrease thick nasal discharge and postnasal drip (*Chiu et al., 2008*). It also found to improve mucociliary clearance as evidence in saccharin transit time (*Isaacs et al., 2011*). An *in vitro* study of *S. aureus* and *P. aeruginosa* biofilm done by *Desrosiers, Myntti & James (2007)*, found that surfactant delivered by hydrodynamic force further enhanced the reduction of biofilm mass.

Additionally, baby shampoo acts as chemical surfactant with antimicrobial properties. It has the ability to disrupt cell membranes, increase membrane permeability—potentially causing metabolite leakage and interfere with membrane functions (*Van Hamme, Singh & Ward, 2006*). Therefore, baby shampoo may inhibit biofilm formation and eradication of planktonic organism. Baby shampoo is a mild surfactant which rinses effectively without causing a burning sensation and generates less foam, making it suitable for nasal irrigation.

## Study limitations and suggestions

The primary limitation of this study is the limited availability of financial resources, which may have influenced methodological design and strategy. Biofilm detection was conducted without CSLM or FISH due to their high cost, although we acknowledge that these methods may enhance biofilm detection. Nevertheless, in our study biofilm was successfully identified using H&E staining and light microscopy, supported by Gram staining. The methods are supported by previous studies, which have demonstrated that biofilm detection using H&E staining is reliable in CRS patients (*Hochstim et al., 2010*; *Hong et al., 2014*; *Tóth et al., 2013*). Furthermore, the methods are easily reproducible and cost effective, making it suitable to be used particularly in centers without CSLM and FISH

services. We selected only the two most commonly found bacteria biofilms in CRSwNP based on previous studies, due to the high cost of ATCC strains. As a result, the reductions of biofilms using diluted 1% baby shampoo of the other biofilms was not investigated.

We suggest using CSLM method with BacLight methods or FISH analysis with species-specific oligonucleotides probes in the next study on biofilm, which is currently most widely used for biofilm detection and to speciate the organisms involved. Sampling methods should be improved by taking samples from all involved paranasal sinuses in view of the reported heterogenicity of organism community in the nasal cavity. We recommend a prospective multi center studies in the future to evaluate the effectiveness of diluted baby shampoo 1% in biofilm reduction comparing populations of urban and rural areas to achieve improved tailored management in our local population.

## CONCLUSIONS

The prevalence of biofilm in patients with CRSwNP in our study is 21.7%. There were significant effects of biofilm reduction following the use of diluted 1% baby shampoo and significant association of *S. aureus* & *P. aeruginosa* with biofilm formation. Our findings suggest a role of screening patients with CRSwNP for biofilms in selecting patients that can benefit from 1% diluted baby shampoo treatment.

### Funding

This work was supported by Faculty of Medicine, Universiti Kebangsaan Malaysia (FF2016369). The funders had no role in study design, data collection and analysis, decision to publish, or preparation of the manuscript.

### Grant Disclosures

The following grant information was disclosed by the authors:
Faculty of Medicine, Universiti Kebangsaan Malaysia: FF2016369.

### Competing Interests

The authors declare there are no competing interests.

### Author Contributions

- Farah Dayana Zahedi conceived and designed the experiments, performed the experiments, analyzed the data, prepared figures and/or tables, authored or reviewed drafts of the article, and approved the final draft.
- Mun Yee Soo conceived and designed the experiments, performed the experiments, analyzed the data, prepared figures and/or tables, authored or reviewed drafts of the article, and approved the final draft.
- Muttaqillah Najihan Abdul Samat conceived and designed the experiments, performed the experiments, authored or reviewed drafts of the article, and approved the final draft.

- Suria Hayati Md Pauzi conceived and designed the experiments, performed the experiments, authored or reviewed drafts of the article, and approved the final draft.
- Salina Husain conceived and designed the experiments, authored or reviewed drafts of the article, and approved the final draft.
- Aneeza Khairiyah Wan Hamizan analyzed the data, authored or reviewed drafts of the article, and approved the final draft.
- Baharudin Abdullah analyzed the data, authored or reviewed drafts of the article, and approved the final draft.
- Balwant Singh Gendeh conceived and designed the experiments, authored or reviewed drafts of the article, and approved the final draft.
- Abdul Samat Ismail analyzed the data, prepared figures and/or tables, and approved the final draft.
- Rocky Vester Richmond analyzed the data, prepared figures and/or tables, authored or reviewed drafts of the article, editing, and approved the final draft.

## Human Ethics

The following information was supplied relating to ethical approvals (i.e., approving body and any reference numbers):

Universiti Kebangsaan Malaysia Research Ethics Committee granted ethical approval to carry out the study within its facilities (JEP2016510)

## Data Availability

The raw data are available in the Supplementary File.

## Supplemental Information

Supplemental information for this article can be found online at http://dx.doi.org/10.7717/peerj.19134#supplemental-information.

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
