# Peer review of "The effect of diluted 1% baby shampoo on biofilm reduction in chronic rhinosinusitis with nasal polyposis"

_PeerJ, doi:10.7717/peerj.19134_

## Round 0.1 · original submission · Major Revisions

Although one of the reviewers recommended rejection, I decided to give you an opportunity to address the issues pointed by the reviewers and to revise your manuscript accordingly.

·

Basic reporting

NO COMMENT

Experimental design

NO COMMMENT

Validity of the findings

THE ARTICLE DOES POSE A NEW FORM OF ASSESSMENT IN MANAGING CRSwP.
THE AIMS AND OBJECTIVES HAVE BEEN STUDIED WELL AND THE CONCLUSION IS ADEQUATE.

Additional comments

Respected Author,

Congratulations on an excellent article.
A couple of changes are needed which has been added to the PDF as annotations.
Please do elaborate on the same.

Regards.

Reviewer 2 ·

Basic reporting

• The paper is clear and comprehensible, though minor grammatical errors (especially tense) should be addressed (e.g., lines 53, 55, 58, 64, 71, 76, 93, 99). A review by a colleague proficient in English or a professional editing service is recommended.
• Minor revisions include:
o Abbreviate "chronic rhinosinusitis with nasal polyps" (e.g., Ln 211) to CRSwNP.
o Italicize S. aureus and P. aeruginosa (e.g., Ln 272, 296, 297).
o Add references in several places (e.g., Ln 61-63, 73-74, 81, 94).
o Ensure consistency in referencing style (e.g., Ln 300 "Bendouah et al., 2006" vs. Ln 295 "Prince et al. reported").
• Tables and figures are relevant, but legends would improve clarity. Ensure consistent use of "1% baby shampoo" throughout.

Experimental design

The research question is well stated and revisited in the Discussion and Conclusion sections.
• Methods are adequately described. However, it would be helpful for the authors to explain the rationale for selecting H&E staining and Gram saining over more advanced techniques like CSLM for biofilm detection? Is this a potential limitation of the study?

Validity of the findings

The authors present data using robust statistical methods and have documented their results comprehensively. Given the rise in antibiotic resistance, their findings support the need for alternative measures to manage CRSwNP. However, the discussion would benefit from a deeper exploration of current treatments, such as corticosteroids, topical antimicrobials, and immunologic biologics, which are commonly used for CRSwNP. Additionally, since this is an in vitro study, the authors should be cautious not to overstate the findings. Future follow-up studies comparing biofilms in vivo would provide more clinically relevant insights.

Additional comments

• The authors mention using normal saline but should discuss other nasal rinses like xylitol or nasal sprays with benzalkonium chloride.
• Clarify if tissue from cystic fibrosis (CF) patients was included, and if so, whether there are differences in biofilms compared to non-CF patients.
• Ln 244-250: The claim that CRS affects quality of life more than COPD and heart disease would benefit from direct comparative citations.
• While the limitations are well noted, the authors could better elaborate on how these limitations might have impacted the results or interpretations. This would significantly strengthen the critique.

Reviewer 3 ·

Basic reporting

The authors of this study aimed to evaluate the effect of 1% baby shampoo on biofilm reduction in chronic rhinosinusitis with nasal polyposis. However, the manuscript lacks scientific rigor, particularly in its methodological approach and preparation. The results are difficult to interpret and follow. I recommend revising the manuscript thoroughly, including the addition of well-designed experiments, to enhance the study's scientific quality and relevance.

Experimental design

- Accuracy in writing: The manuscript lacks precision in formatting bacterial names. The full name of bacteria should be provided on first mention, with abbreviations used thereafter. Additionally, bacterial names must be consistently italicized throughout the text.
- Bacterial isolation details: The bacterial isolation process is insufficiently described. The authors should confirm bacterial strains using biochemical tests and 16S rRNA sequencing. It’s likely that a greater variety of isolates could be obtained from patients; the manuscript should include a detailed list of isolated bacteria, corresponding with patient data.
- Limited number of clinical isolates: The study uses a low number of clinical isolates, limiting the generalizability of the results.
- Biofilm incubation duration: The authors do not justify the choice of an 8-day biofilm incubation period. Biofilm formation involves attachment, maturation, and detachment phases; it’s unclear if this incubation time appropriately represents these stages.
- Selection of bacterial isolates for Testing: It is unclear which bacterial isolates were used to test the 1% baby shampoo. Although the manuscript suggests two isolates were studied, this is not evident in Table 3.
- Biofilm-forming capacity: There is no classification or analysis of the biofilm-forming capacity of the isolated bacteria, which is an essential detail missing from the results section.
- Relevance of results: The effectiveness of 1% baby shampoo is presented ambiguously. The results should specify which bacterial isolates were used to generate the data.
- Result presentation: The results are challenging to follow. Each section of the results should explicitly reference the relevant tables and figures to enhance clarity and reader understanding.

Validity of the findings

The novelty of this manuscript is relatively low, as previous studies have already demonstrated the effects of 1% baby shampoo on biofilm reduction. The discussion section lacks an explanation of the mechanism by which 1% baby shampoo reduces biofilm.

Additional comments

The limitations of this manuscript include a small number of bacterial isolates, inaccuracies in manuscript preparation (notably errors in bacterial nomenclature and spacing issues), and imprecision in the biofilm reduction method. To enhance the manuscript's impact, I recommend addressing these issues comprehensively. Improving the originality of the study, possibly by incorporating novel experimental techniques or investigating unique mechanisms of action, would significantly strengthen its scientific value. A more rigorous approach to method accuracy and manuscript formatting will also contribute to higher credibility and clarity.

---

## Round 0.2 · accepted · Accept

It seems that the constructive critiques of the reviewers were adequately addressed. Therefore, revised manuscript is acceptable now.

·

Basic reporting

no comment

Experimental design

no comment

Validity of the findings

no comment

Additional comments

The revised text is good and has an air-tight protocol.